# Conversion of the Native N-Terminal Domain of TDP-43 into a Monomeric Alternative Fold with Lower Aggregation Propensity

**DOI:** 10.3390/molecules27134309

**Published:** 2022-07-05

**Authors:** Matteo Moretti, Isabella Marzi, Cristina Cantarutti, Mirella Vivoli Vega, Walter Mandaliti, Maria Chiara Mimmi, Francesco Bemporad, Alessandra Corazza, Fabrizio Chiti

**Affiliations:** 1Department of Experimental and Clinical Biomedical Sciences “Mario Serio”, University of Florence, 50134 Florence, Italy; matteomoretti109@gmail.com (M.M.); isabella.marzi@unifi.it (I.M.); francesco.bemporad@unifi.it (F.B.); 2Department of Medical Area, University of Udine, 33100 Udine, Italy; cristina.cantarutti@uniud.it (C.C.); walter.mandaliti@gmail.com (W.M.); alessandra.corazza@uniud.it (A.C.); 3School of Biochemistry, University of Bristol, Bristol BS8 1TD, UK; mirella.vivolivega@bristol.ac.uk; 4Department of Molecular Medicine, University of Pavia, 27100 Pavia, Italy; chiara.mimmi@unipv.it; 5INBB, Viale Medaglie d’Oro 305, 00136 Roma, Italy

**Keywords:** neurodegeneration, self-assembly, oligomerisation, native-like, micelle

## Abstract

TAR DNA-binding protein 43 (TDP-43) forms intraneuronal cytoplasmic inclusions associated with amyotrophic lateral sclerosis and ubiquitin-positive frontotemporal lobar degeneration. Its N-terminal domain (NTD) can dimerise/oligomerise with the head-to-tail arrangement, which is essential for function but also favours liquid-liquid phase separation and inclusion formation of full-length TDP-43. Using various biophysical approaches, we identified an alternative conformational state of NTD in the presence of Sulfobetaine 3-10 (SB3-10), with higher content of α-helical structure and tryptophan solvent exposure. NMR shows a highly mobile structure, with partially folded regions and β-sheet content decrease, with a concomitant increase of α-helical structure. It is monomeric and reverts to native oligomeric NTD upon SB3-10 dilution. The equilibrium GdnHCl-induced denaturation shows a cooperative folding and a somewhat lower conformational stability. When the aggregation processes were compared with and without pre-incubation with SB3-10, but at the identical final SB3-10 concentration, a slower aggregation was found in the former case, despite the reversible attainment of the native conformation in both cases. This was attributed to protein monomerization and oligomeric seeds disruption by the conditions promoting the alternative conformation. Overall, the results show a high plasticity of TDP-43 NTD and identify strategies to monomerise TDP-43 NTD for methodological and biomedical applications.

## 1. Introduction

Amyotrophic lateral sclerosis (ALS) is a fatal disorder characterized by a progressive degeneration of the upper and lower motor neurons of the brain, brainstem, and spinal cord, but also in the frontotemporal cortex and hippocampus in a fraction of patients [1,2,3], leading to muscle weakness, wasting and spasticity [4]. In 2006, the transactive response (TAR) DNA-binding protein (TDP-43) was identified as the main component of intraneuronal cytosolic ubiquitinated protein aggregates found in ALS and tau-negative, ubiquitin-positive frontotemporal lobar degeneration (FTLD) patients [1,2]. Numerous post-translational modifications of TDP-43 have been described in the aggregates, such as abnormal polyubiquitination and hyperphosphorylation as well as partial proteolysis to form C-terminal fragments [1,2,5,6].

The TDP-43 protein contains 414 amino acid residues and the encoding gene *TARDBP* is located on chromosome 1. It comprises an N-terminal domain (NTD, aa 1–76), a nuclear localization signal (NLS, aa 82–98), two RNA recognition motifs (RRM1, aa 106–176, and RRM2, aa 191–259), a nuclear export signal (NES, aa 239–250), and a C-terminal domain (CTD, aa 274–414), which encompasses a prion-like glutamine/asparagine-rich (Q/N) region (aa 345–366) and a glycine-rich region (aa 366–414) [7,8,9,10,11,12]. TDP-43 was first proposed to be natively dimeric or at least to exist in a monomer-dimer equilibrium under normal physiological conditions [13,14,15,16]. The dimerisation of TDP-43 was shown to occur through interactions of the N-terminal domains, and these interactions were proposed to be responsible for the dimerisation of the entire full-length protein [14,15,16]. It was also found that the deletion of the N terminus, or even the first nine residues, is sufficient to abolish NTD folding, dimerisation and the TDP-43–regulated RNA splicing [15]. With the elucidation of the monomeric structure of NTD with NMR spectroscopy, models of the dimer were proposed [12,17,18]. Later on, the NTD and full-length TDP-43 were found to form oligomers beyond the dimer and that TDP-43 oligomerisation was necessary for its function as mutants destabilising the NTD-NTD interface without affecting the NTD folded structure inhibit the splicing activity of full-length TDP-43 expressed in cells [19,20].

Although NTD-mediated TDP-43 dimerisation/oligomerisation appears to be essential for TDP-43 function [14,15,19,20], extensive NTD-mediated TDP-43 oligomerisation was also found to enhance the liquid-liquid phase separation (LLPS) at physiological concentrations [20,21,22], as well as formation of solid phase cytotoxic inclusions in the cytoplasm [15,23].

The NTD monomer consists of seven-eight β-strands and a single α-helix arranged in an axin-1 DIX domain fold [12,17,18,19,20], which facilitates dimer formation and further oligomerisation by a head-to-tail disposition of the individual subunits [19,20,24]. The conformational stability of the domain, as measured by the hydrogen/deuterium exchange at pH 3.8 and 25 °C, was found to be 15.9 ± 0.5 kJ mol^−1^ [12]. Using equilibrium urea denaturation curves monitored with intrinsic fluorescence and far-UV circular dichroism at pH 7.4 and 25 °C, higher conformational stability values of 20.1 ± 1.5 kJ mol^−1^ and 21.8 ± 1.5 kJ mol^−1^ were found, respectively [24], most probably due to the neutral pH of the measurements. The β-strands, α-helix, and three of four turns are rigid, although the loop formed by residues 47–53 appeared mobile [17]. All X-Pro peptide bonds adopt a trans configuration and the two cysteine residues at positions 39 and 50 are reduced and distantly separated on the surface of the protein [12,17,19,20].

The NTD of TDP-43 follows a folding process characterized by a series of steps [24]. The NTD unfolded state (U) consists of a pre-equilibrium between molecules having all the X-Pro peptide bonds in a native trans configuration and others with one or more X-Pro peptide bonds in a non-native cis configuration. The unfolded state U converts into a collapsed state (CS), which then converts into the fully folded monomer (F) following two parallel pathways: in the first CS converts directly to F, whereas in the second, CS converts into an intermediate state (I) that is on-pathway to F. All these U, CS and I states maintain the partition equilibrium between cis and trans-X-Pro peptide bonds, but folding can occur only from molecules having all-trans X-Pro bonds. Once the fully folded state F is attained, the NTD is able to dimerise into a head-to-tail homodimer that presents the further ability to form higher molecular weight assemblies [24]. Following urea denaturation at equilibrium, a native-like dimeric state was also detected at low urea concentrations (F*). Moreover, the various NMR studies of TDP-43 NTD show spectral differences in the various experimental conditions in which the protein domain is folded [10,12,14,17,18,20]. Overall, these structural and folding studies have revealed that TDP-43 NTD is a highly plastic protein able to adopt different conformational states.

In this work, we purified the NTD of TDP-43 and identified conditions to obtain two conformational states of the protein—one that is fully folded, native and dimeric and another, obtained through the use of a zwitterionic detergent Sulfobetaine 3-10 (SB3-10), that presents a monomeric, alternative, non-native, partially folded structure. We will show that the SB3-10-stabilised conformational state of TDP-43 NTD retains a cooperative fold and high conformational stability in spite of a remarkably different three-dimensional structure but does not retain the ability to dimerise and has a remarkably lower propensity to oligomerise and aggregate.

## 2. Results

### 2.1. The Alternative NTD Conformation Is Enriched with α-Helical Structure and Less Densely Packed

The NTD of TDP-43, characterised by a 6x His-tag followed by TEV protease cleavage site and the first 77 residues of TDP-43, was purified and found to be electrophoretically pure. Initially, the structure and oligomerisation of the protein domain were studied in 5 mM sodium phosphate buffer, 50 mM NaCl, 1 mM dithiothreitol (DTT), pH 7.4, in the absence or presence of 3.0% (*w*/*v*) SB3-10, at 25 °C. This zwitterionic detergent was initially used to stabilize the protein domain during purification, as it is normally used in protein purification protocols. However, we soon realized that it led to a different conformational state of the protein domain and we, therefore, decided to use it more systematically to investigate the structural plasticity of the TDP-43 NTD.

The far UV circular dichroism (CD) spectrum for the protein domain without SB3-10 featured two negative peaks at ca. 195 and 209 nm with a distinct small positive band at 233 nm (Figure 1A). The obtained CD spectrum was very similar to the other TDP-43 NTD spectra reported in the literature [14,18,24]. It was also found to be stable within 24 h, as ten CD spectra recorded at regular time intervals within this timeframe were superimposable and led to similar estimates of secondary structure types (Figure 1A, inset) when analysed with the BeStSel algorithm, which takes into account sheet twisting as a deconvolution parameter [25]. The secondary structure distribution is also in agreement with the values deduced from the X-ray and NMR structures of this protein domain [19,20]. The CD spectrum of the protein domain with 3.0% SB3-10, on the other hand, exhibits two negative peaks at ca. 207 nm and 224 nm and a positive peak at ca. 192 nm, thus showing a higher α-helical content (Figure 1A). Similar, to the spectrum in 0.0% SB3-10, it was found to be stable within 24 h but led to a different distribution of secondary structure types when analysed with the BeStSel algorithm, with a higher content of α-helical structure at the expense of the β-sheet structure, and a higher quantity of turns (*p* < 0.001, Figure 1A, inset).

The intrinsic fluorescence spectrum of TDP-43 NTD in the absence of SB3-10 showed a single peak at 319 nm, indicating a fully folded structure, with the Trp68 residue well buried in the hydrophobic core of the protein (Figure 1B). In 3.0% SB3-10, the intrinsic fluorescence increased in intensity and underwent a red-shift to 339 nm, showing a Trp68 residue more exposed to the solvent and a partially folded structure (Figure 1B). The degree of solvent exposure of the Trp68 residue of TDP-43 NTD was also studied using a Stern–Volmer assay with acrylamide as a quencher (Figure 1C). TDP-43 NTD treated with 0.0% SB3-10 showed a linear plot of *F*_0_/*F* versus acrylamide concentration with a *K_SV_* value of 8.8 ± 0.3 M^−1^, while TDP-43 NTD treated with 3.0% SB3-10 showed a non-linear plot, with a *K_SV_* value of 14.3 ± 0.6 M^−1^ and a K_ST_ value of 4.5 ± 0.2 M^−1^ (Figure 1C). The higher *K_SV_* value (*p* < 0.01) and the presence of a static quenching component (K_ST_) indicate a greater exposure of the Trp68 residue to the solvent for the protein domain in 3.0% SB3-10.

### 2.2. The Alternative NTD Conformation Studied with NMR Is Highly Dynamic

We then performed an NMR analysis on the ^15^N and ^13^C labelled protein domains in both conditions. The ^1^H-^15^N HSQC spectrum in 3% (*w*/*v*) SB3-10 shows a low degree of resonance spreading, typical of 〈-helix rich and partially folded proteins (Figure 2A), in contrast to the HSQC spectrum of the native NTD in 0% SB3-10 previously shown [17,18,20] and here analysed as a control (Appendix A). At 25 °C, 84 backbone amide peaks can be observed in the HSQC spectrum in 3% SB3-10 and at 17 °C, this number raises to 99 (Appendix A). Considering that the NTD has 72 non-proline residues, we infer that some residues exhibit multiple forms. Seven backbone peaks could not be unambiguously assigned (E21, R44, N45, C50, M51, R55 and L56). Thirteen alternative forms were assigned. Four of them were found to be related to C39 oxidation (A33, A38, L41, R42) because DTT addition led to a decrease in the peak intensities of the second form (Appendix A). Cα/β chemical shift values of C39 are however compatible with the absence of a disulphide bond, indicating a different type of thiol oxidation. The other nine residues were not affected by the addition of DTT and are residues 24–29, 59–60 and 63 that form a well-defined cluster in the native structure (Appendix A), indicating structural mobility at these sites. In particular, three forms were detected for A63. Spectra acquired at low temperature (17 °C) show that one of the two conformations (hereinafter referred to as form A) is poorly populated relative to form B, whereas it becomes more populated as the temperature increases suggesting an equilibrium between the two species. ^1^H, ^13^C and ^15^N chemical shifts have been deposited in the BioMagResBank (http://www.bmrb.wisc.edu, accessed on 14 June 2022) under the BMRB accession number 51492.

Based on chemical shift values, we assessed the secondary structure composition by Talos+ [26], chemical shift index (CSI) [27], and Secondary Structure Propensity (SSP) [28]. All methods confirmed, in agreement with the CD analysis, the increase in α-helical structure and the concomitant decrease in β-sheet structure. In addition to the 28-33 helix, which adopts a helical structure in native NTD as well, other two non-native helices were found in the N-terminal and C-terminal domains (Appendix A, Figure 2B and Appendix A). Moreover, form A was found to have a longer helix 26–33, starting from residue 26 rather than residue 28. A consensus prediction of extended conformation for residues 16–18, the central portion of ^®^-strand 2 in native NTD, was found for the three predictive methods. In native NTD β-strand 2 forms an antiparallel β-sheet with ^®^-strand 1 that, according to CSI (Appendix A) and SSP analysis (Appendix A), also shows a propensity for ^®^-strand conformation, although with a low score, in agreement with the little antiparallel β-sheet structure observed with far-UV CD (10.6 ± 2.3%). Talos+ analysis indicates that those strands of the protein are highly dynamic with random coil indexes (RCI) below or equal to 0.7 (Figure 2C).

We also recorded ^1^H-^15^N HSQC spectra at different temperature values from 17 to 39 °C (Appendix A). The slope of the variation of the peak chemical shift as a function of temperature reveals whether or not the amide group is involved in a hydrogen bond; in particular, a slope < −4.5 ppb/K reveals a hydrogen bond [29]. Because of the overlap of some peaks, it was not possible to extract this information for all the assigned residues. The hydrogen bond pattern is largely consistent with the secondary structure obtained from CSI and Talos+ analysis (Appendix A).

The nature of this alternative TDP-43 NTD conformation in 3% (*w*/*v*) SB3-10 was also studied by ^15^N{^1^H} NOE experiments [30]. The average NOE value calculated over the sequence is quite low (0.42 ± 0.18, maximum value 0.69) and the predicted structured regions also show NOE values lower than expected (Figure 2D), confirming the overall dynamic nature of this conformation. Furthermore, form B presents slightly lower NOE values. Experimental NOEs and predicted RCIs are in agreement and report reduced flexibility in the structured regions (at the termini and the central helix) and high mobility in the regions encompassing residues 6–19 and 46–67. By decreasing the temperature from 25 to 17 °C, we observed a remarkable increase in the NOE values in the 39–42 region (Appendix A). This region lacks a secondary structure, as inferred from NMR chemical shifts, but in the native conformation, it has an extended secondary structure (Appendix A and Figure 2B). Residues R6 and V57 also decrease significantly their flexibility at 17 °C and both are involved in the formation of β-strands in the native conformation.

### 2.3. The Alternative NTD Conformation Is a Monomer

The hydrodynamic diameters of both TDP-43 NTD conformations were determined using dynamic light scattering (DLS). The distribution of light scattering intensity vs. the apparent hydrodynamic diameter revealed that TDP-43 NTD in the absence of SB3-10 has a monodispersed distribution with a hydrodynamic diameter of 5.1 ± 0.3 nm, while in the presence of 3.0% SB3-10, the hydrodynamic diameter was 4.7 ± 0.2 nm (Figure 3A). The hydrodynamic diameters expected for the TDP-43 NTD devoid of any tag are 3.23 and 4.28 nm for folded monomer and dimer, respectively [24]. Since our TDP-43 NTD has an additional unfolded tag, it can be concluded that native TDP-43 NTD in 0.0% SB3-10 occurs as a folded dimer, possibly adopting a higher-order oligomer. By contrast, the decrease of the hydrodynamic diameter of the protein domain from 5.1 ± 0.3 nm to 4.7 ± 0.2 nm upon SB3-10 addition, concomitantly with its partial unfolding suggested by the CD, fluorescence and NMR analyses reported above, suggests that the protein domain is a partially folded monomer in 3.0% SB3-10.

The two different conformations were then studied using analytical SEC. The elution volume (V_e_) peak was 16.7 ± 0.3 mL in 0.0% SB3-10 (Figure 3B, blue). Through the use of the calibration curve obtained with folded proteins of known mass (Figure 3B, inset, blue), this V_e_ was found to correspond to a molecular mass of 20.2 ± 2.6 kDa, closer to that of a dimer rather than a monomer (expected molecular weights of 22.2 and 11.1 kDa, respectively). This is in agreement with previous SEC studies [24]. The alternative form in 3.0% SB3-10 eluted as a major peak at 16.0 ± 0.3 mL preceded by a minor one at 14.9 ± 0.3 mL (Figure 3B, red). Using the same analysis with a calibration curve corrected for partially unfolded (pre-molten globule) standard proteins (Figure 3B, inset, red), a molecular weight of 11.4 ± 1.7 kDa was obtained for the major peak, confirming its monomeric state.

Lastly, Förster Resonance Energy Transfer (FRET) was used to further investigate the conformational state of the TDP-43 NTD in 3.0% SB3-10. Since the NTD has two cysteine residues at position 39 and 50, a mutant that maintained only Cys39 was purified, namely C50S. This C50S mutant was labelled with the thiol-reactive probes 5-((((2-iodoacetyl)amino)ethyl)amino)naphthalene-1-sulfonic acid (1,5-IAEDANS) as a donor (D) and 6-iodoacetamidofluorescein (6-IAF) as an acceptor (A). FRET analysis was then performed acquiring the fluorescence spectra of the samples containing only NTD-D, only NTD-A and both NTD-D and NTD-A at a 1:1 molar ratio (Figure 3C, dashed, dotted and continuous red lines, respectively). The spectrum obtained from the algebraic sum of the first two spectra (Figure 3C, mixed dotted/dashed orange line) was found to be superimposable to the third spectrum (Figure 3C, continuous red line). Hence, unlike the TDP-43 NTD head-to-tail dimer previously studied in 0.0% SB3-10, in which significant FRET was observed [24], in the presence of 3.0% SB3-10 FRET was not observed, further confirming the monomeric state of the alternative TDP-43 NTD conformation.

### 2.4. Alternative NTD Exhibits a Lower Cooperativity Than Native NTD

The conformational stabilities of the two forms of TDP-43 NTD were analysed by means of intrinsic tryptophan fluorescence and using GdnHCl as a chemical denaturant (Figure 4). 30 samples containing TDP-43 NTD and GdnHCl concentrations ranging from 0.0 to 5.0 M were prepared and their tryptophan emission spectra were recorded. The analysis was then repeated adding 3.0% SB3-10 in all samples. Afterwards, the ratio between the fluorescence values in two distinct regions of wavelength was measured for each spectrum and the obtained ratio was plotted as a function of the molar concentration of GdnHCl (Figure 4). In both cases, the GdnHCl-induced denaturation of TDP-43 NTD follows a two-state equilibrium and can be fitted to the model edited by Santoro and Bolen [31]. The results of the analysis with 0.0% SB3-10 yielded values of 22.3±1.4 kJ/mol, 8.7 ± 0.6 kJ/(mol M) and 2.6 ± 0.2 M for $\Delta G_{U-F}^{H20}$, m_eq_ and C_m_, respectively (Figure 4A). The $\Delta G_{U-F}^{H20}$ value is identical, within experimental error (*p* > 0.05), to that obtained previously for the same protein domain using urea as a chemical denaturant, i.e., 21.8 ± 1.5 kJ/mol [24]. For TDP-43 NTD treated with 3.0% SB3-10, values of 19.2 ± 3.3 kJ/mol, 5.7 ± 1.2 kJ/(mol M) and 3.4 ± 0.2 M were obtained for the same parameters, respectively (Figure 4B). The two sets of values indicate a cooperative transition in both cases, with significantly lower cooperativity for the alternative conformation in 3.0% SB3-10, as indicated by a m_eq_ value decreased by 35 ± 15% (*p* < 0.05) relative to that measured in its absence. This reduction indicates an increase in the solvent-exposed surface area. The overall stability is also lower with the $\Delta G_{U-F}^{H20}$ value decreasing by ca. 3 kJ/mol, although the difference does not reach statistical significance in this case.

### 2.5. The Native-to-Alternative NTD Transition Is Reversible and Associated with SB3-10 Micelle Formation

In order to investigate the existence and reversibility of a possible transition between the two conformations in 0.0% and 3.0% SB3-10, intrinsic fluorescence spectra were recorded treating TDP-43 NTD with increasing concentrations of SB3-10 from 0.0% to 3.0%. The data show that starting from a concentration of ca. 0.6% SB3-10 there is a sharp increase in the fluorescence intensity up to ca. 1.4% (Figure 5A). It is also possible to see a red-shift of the wavelength of maximum fluorescence from 319 nm to 339 nm, within the same range of SB3-10 concentration (Figure 5A). The centre of spectral mass (COM) for each spectrum was calculated and the COM was plotted vs. SB3-10 percent concentration (Figure 5B). The SB3-10-induced conformational change of TDP-43 NTD follows an apparent two-state equilibrium and can be fitted to the model edited by Santoro and Bolen [31]. The results of this analysis yielded values of 16.0 ± 1.6 kJ/mol, 16.2 ± 1.4 kJ/(mol M) and 0.98% (*w*/*v*) for $\Delta G_{U-F}^{H20}$, m_eq_ and C_m_, respectively. Spectra recorded at 0.6% SB3-10 at the same protein concentration, before and after the protein was incubated at 1.8% SB31-0 for 1 h, were essentially superimposable indicating the reversibility of the transition.

In order to explore if the conformational transition observed for TDP-43 NTD upon addition of SB3-10 was due to the formation of micelles by this compound, the solution containing 3.0% SB3-10 without TDP-43 NTD was analysed using DLS (Figure 5C). The distribution of light scattering intensity vs. the apparent hydrodynamic diameter revealed that at 3.0% SB3-10 the formation of micelles was evident, with a monodispersed distribution and a hydrodynamic diameter of 3.87 ± 0.05 nm. The overall intensity of scattered light was one order of magnitude lower than that observed when the protein was present under the same solution conditions, ruling out that the size distribution reported for the protein in Figure 3A is dominated by SB3-10. The SB3-10 concentration-dependent formation of the micelles was then studied by monitoring the light scattering intensity of the solution with an increasing concentration of SB3-10 in the absence of protein (Figure 5D). The distribution of light scattering intensity vs. SB3-10 percent concentration revealed that micelles begin to form starting from a concentration of 0.6% and then increase stabilizing at 1.4%, with a midpoint around 1.0%. It therefore appears that the SB3-10-induced conformational transition of TDP-43 NTD monitored with intrinsic fluorescence and the formation of SB3-10 micelles monitored with light scattering are fully superimposable, indicating that the conformational transition of this protein domain is induced by the formation of SB3-10 micelles.

### 2.6. Alternative NTD Has a Lower Propensity to Aggregate

Native NTD is known to oligomerise in a head-to-tail fashion at sufficiently high concentrations, well beyond the dimer [19,20]. Such oligomers represent a functional state of the NTD [19,20], but extensive and uncontrolled oligomerisation leads liquid–liquid phase separation and possibly inclusion formation of the full-length protein [20,22]. Using analytical ultracentrifugation (AUC), an isodesmic dissociation constant of 40 µM was found for the native NTD oligomers [20]. In order to investigate the oligomerisation process of TDP-43 NTD in the absence of SB3-10, NTD was incubated at a concentration of 0.5 mg/mL (45 μM), in 5 mM sodium phosphate buffer, 50 mM NaCl, 1 mM DTT, pH 7.4, 25 °C, which are solution conditions known to promote oligomerisation [20,32].

Under these conditions, the size distribution by light scattering intensity showed initially (0 h) a significant amount of dimeric or low molecular weight oligomeric state but also high molecular weight species (Figure 6A,E). Since light scattering intensity scales with the square of the mass, the latter species are initially quantitatively negligible. However, prolonged incubation leads to the slow decay of the Gaussian curve associated with the dimer/oligomer, until it disappears after 3 h (Figure 6A,E). A similar decay was observed in the presence of small concentrations of SB3-10, up to 0.6%, that is in the pre-transition region of the conversion between the two conformational states studied here, when the protein domain maintains initially its native folded state (Figure 6B,F). The apparent rate constant of decay of the DLS peak did not change significantly with SB3-10 concentration (Figure 6I, r = 0.433, *p* > 0.05), nor was the light scattering intensity associated with this species at time 0 h found to correlate significantly with SB3-10 concentration (Figure 6I, r = 0.303, *p* > 0.05).

In another experiment, NTD was treated differently: starting from the native conformation in 0.0% SB3-10, the protein domain was brought to 1.8% SB3-10 for 1 h, i.e., in the post-transition region in which it adopts the alternative monomeric conformational state, and then diluted down to 0.6% SB3-10, under final protein concentration and solution conditions identical to those of the previous experiments (Figure 6C,G). In this case, the decay of light scattering associated with the low molecular weight peak was slower, with a rate constant 4-fold lower than that observed under identical final conditions but without the intermediate incubation in 1.8% SB3-10 for 1 h (Figure 6I). The rate constants values were indeed 1.03 ± 0.19 h^−1^ and 0.24 ± 0.04 h^−1^, without and with preincubation in 1.8% SB3-10, respectively (*p* < 0.01). In other words, the NTD that adopted transiently the alternative monomeric conformation in 1.8% SB3-10 was found to aggregate, in 0.6% SB3-10, more slowly than the NTD brought to 0.6% SB3-10 directly, in spite of otherwise identical final conditions.

In the presence of 3.0% SB3-10, i.e., in the post-transition region when the protein adopts the alternative monomeric conformational state, the size distribution was dominated at time 0 h by a single monomeric peak, accounting for ca. 92% of light scattering intensity, which then disappeared slowly (Figure 6D,H). The decay occurred in this case with an apparent rate constant 4-fold lower (0.25 ± 0.04 h^−1^) than that observed in 0.0–0.6% SB3-10 (1.03 ± 0.19 h^−1^, *p* < 0.01), but apparently identical to that observed in 0.6% SB3-10 with intermediate incubation in 1.8% SB3-10 (0.24 ± 0.04 h^−1^, *p* > 0.05) (Figure 6I).

### 2.7. Aggregation of Either NTD Conformation Is Not Associated with Detectable Structural Variation

In order to monitor the occurrence of possible structural changes of NTD during the time-dependent self-assembly of the dimer or low molecular weight oligomers monitored with DLS at low SB3-10 concentrations (0.0–0.6%) or of the monomeric alternative conformational state at higher SB3-10 concentrations (>1.6%) we recorded far-UV CD and intrinsic fluorescence spectra of the same protein samples analysed with DLS (same protein concentrations, solution conditions, time points).

The far-UV CD spectra obtained in 0.0%, 0.2% and 0.6% SB3-10 did not show significant variations with time; in agreement, intrinsic fluorescence did not change with time within this range of concentrations, maintaining the characteristic peaks of the native fully folded state (Appendix A), thus indicating a native-like oligomerisation. Similarly, the far-UV CD spectra obtained in 1.8% and 3.0% SB3-10 did not show detectable changes with time, maintaining the characteristic peaks of the alternative partially folded state (Appendix A). In 0.6% SB3-10 the spectra were similar without (Appendix A) or with (Appendix A) the intermediate 1 h incubation in 1.8% SB3-10, indicating the reversibility of secondary structure changes undergone by the protein.

The intrinsic fluorescence spectra in 0.0%, 0.2% and 0.6% SB3-10 maintained a λ_max_ value at ca. 319 nm during self-assembly, indicating the maintenance of the overall exposure of Trp68, typical of the fully folded state (Appendix A). Similarly, the spectra in 1.8% and 3.0% SB3-10 maintained the λ_max_ value at ca. 339 nm during aggregation, which is typical of the alternative conformational state, indicating again a process of self-assembly in which the protein molecules maintained a similar exposure of Trp68 (Appendix A). Similar to the CD analysis, in 0.6% SB3-10 the spectra were similar without (Appendix A) or with (Appendix A) the intermediate 1-h incubation in 1.8% SB3-10. A slight progressive decrease in intrinsic fluorescence was observed under all the conditions studied, attributable to the increasing light scattering as larger assemblies accumulate.

### 2.8. NTD Aggregation Does Not Lead to Formation of Cross-β Structure

We then assayed the ability of the TDP-43 NTD samples, obtained under the various conditions tested, to bind the ThT dye and increase its fluorescence at 485 nm, which is typical of amyloid protein aggregates. The sample in 0.0% SB3-10 was found to have a fluorescence value F identical to that of free ThT in the absence of protein *F*_0_ (Figure 7). A very small increase in ThT fluorescence F relative to the *F*_0_ value was observed in all the remaining conditions after 6 h, but the increase (*F*_0_/*F*) was generally lower than 1.25 (Figure 7). The samples in 3.0% SB3-10 aged beyond 6 h were found to increase the ThT fluorescence to higher extents, typically between 1.25 and 1.4 (Figure 7). This is in sharp contrast with the over 5-fold fluorescence increase expected for amyloid [33,34], ruling out the formation of amyloid-like species for TDP-43 NTD under these experimental conditions.

## 3. Discussion

The NTD of TDP-43 is able to dimerise with a head-to-tail arrangement into a dimeric structure that has been solved by both NMR and X-ray crystallography [19,20]. Following the head-to-tail interaction, this process does not terminate at the dimeric level but proceeds to form larger oligomers [19,20]. The role of NTD dimerisation and oligomerisation in the aggregation process of full-length TDP-43 is still discussed. It has been proposed to be responsible for the oligomerisation of the entire full-length protein [12,14,15,16,17,18,19,20]. Oligomerisation of full-length TDP-43 mediated by the NTD is essential for TDP-43 function [14,15,19,20], but also favours liquid–liquid phase separation and solid-phase inclusion formation of the full-length protein [20,21,22]. The structural plasticity of the TDP-43 NTD resulting in its in dimerisation/oligomerisation is also observed in its folding process from a fully unfolded state, in which the protein domain forms a number of partially folded states before achieving the fully folded dimeric structure, and even populates, at low denaturant concentrations, a native-like dimeric state distinct from the fully native dimeric conformation [24]. Such conformational heterogeneity is also witnessed by NMR spectral differences in the various experimental conditions in which the protein domain is folded [10,12,14,17,18,20].

The use of very small SB3-10 percent concentrations, typically in the range of 1.4–3.0% (*w*/*v*), has allowed us to isolate a stable and well-defined alternative conformation. While the native conformation in the absence of SB3-10 presents a far-UV CD spectrum comparable to those reported in the literature [14,18,24], the alternative one has a different spectrum characterized by a higher content of α-helical structure, a lower content of β-sheet structure and a slightly higher content of turns. NMR spectroscopy confirmed these observations (discussed below). The intrinsic fluorescence spectrum and Stern–Volmer assay also indicate that the Trp68 residue is more exposed to the solvent in this new conformational state relative to the native structure. The DLS, SEC and FRET analyses all indicate that the NTD in 3.0% SB3-10 is a monomer, unlike the native dimer populated in the absence of this detergent but under identical conditions in terms of protein concentration and solution conditions.

The NMR spectral properties of the SB3-10 dependent conformational state differ from those previously observed for the native NTD, as it clearly appears from the low resonance spreading of its HSQC spectrum. In particular, the NMR characterisation indicates the presence of a partially unfolded state together with the conservation of the native α-helix and of the native antiparallel β-sheet contributed by strands 1-2, although the latter is remarkably more dynamic than the native one (Figure 8A,B). On the basis of chemical shift values, two other non-native α-helices located in the N- and C-termini are detected encompassing residues 1–4 and 68–77, respectively (Figure 8A,B). Moreover, the first part of the central native helix along with the preceding β-strand (residues 24–29) and other spatially close residues that in the native structure faces this portion (residues 59,60,63) form a cluster that is present in two different forms in equilibrium, named here A and B (Figure 8A,B). The intensity of the two forms is temperature dependent, suggesting that the two forms might be related to two conformers with different compactness. For some residues, the two forms show an NH combined chemical shift difference quite marked (between 0.25 and 0.94 ppm) indicating a different chemical environment that reflects distinct conformations and also indicates structure formation under these conditions for at least one form.

The alternative conformation adopted in 3.0% (*w*/*v*) SB3-10 is able to return to its initial native conformation if diluted to a smaller concentration of detergent, conditions in which the native conformation is stable, showing how this process is fully reversible. Its conformational stability ($\Delta G_{U-F}^{H20}$) in 3.0% SB3-10 is only approximately 3 kJ/mol lower than that of the native conformation in 0.0% SB3-10, as determined with GdnHCl-induced equilibrium denaturation. The *m_eq_* value, which reports on the cooperativity of the GdnHCl-induced unfolding transition and on the change of solvent-exposed surface area (∆ASA) upon unfolding, is 35 ± 15% lower than that of the native state but is sufficiently high to indicate a cooperative transition. The lower *m_eq_* value suggests, however, significantly lower cooperativity, possibly due to the increase of the surface area exposed to the solvent in the folded alternative conformation compared to the folded native conformation.

In our time-dependent aggregation assays, we observed a slow, yet detectable, aggregation in the sample left untreated with SB3-10 and in those samples in which the concentration of the detergent was kept low. In both cases, the disappearance of the low molecular weight species (either monomeric, dimeric or oligomeric) was not accompanied by detectable changes in the far-UV CD and intrinsic fluorescence spectra, indicating the maintenance of the initial conformational states in both cases and further supporting the notion that TDP-43 NTD self-assembly consists in a native-like aggregation process similar to that observed for other proteins [35,36,37,38,39]. Aggregation is slower in 3.0% (*w*/*v*) SB3-10, but this effect cannot be attributed necessarily to the SB3-10-mediated conformational change of the TDP-43 NTD, as it may well arise from the effect of the detergent in the formation of intermolecular interactions. However, when the aggregation processes were compared with and without pre-incubation with 1.8% (*w*/*v*) SB3-10, but with an identical final SB3-10 concentration of 0.6% (*w*/*v*) promoting the native state, a slower aggregation was observed in the NTD sample pre-incubated in SB3-10, in spite of the rapid reversibility of the conformational change and the native conformation adopted by the protein domain in both cases. This can be attributed to the ability of the condition promoting temporarily the alternative conformation to monomerise the protein and disrupt any intermolecular interaction (Figure 8C). Under physiological conditions, the native monomers, dimers and oligomers exist initially in a pre-equilibrium where the mutual conversions are slow (Figure 8, left). This condition is favourable for aggregation as dimers and oligomers are present already. The alternative state has a very low propensity to oligomerise (Figure 8C, centre), but the lower propensity of the native state to oligomerise and self-assemble is maintained when the alternative state is re-located under native conditions to form the native state (Figure 8C, right) because the conversion to dimers and oligomers is slow.

Under conditions in which the alternative conformational state was observed (3.0% SB3-10), the critical micellar concentration (CMC) of the detergent was largely exceeded, leading to hypothesize that the NTD conformational change was induced by the newly formed SB3-10 micelles. The SB3-10 titration of the TDP-43 NTD structural change monitored spectroscopically, on the one hand, and of the SB3-10 micelle formation monitored with light scattering intensity, on the other hand, indicate that the two transitions coincide as they start and end at identical SB3-10 concentrations and also have similar transition midpoints. This indicates that the structural change of TDP-43 NTD is driven by SB3-10 micelle formation. Therefore, the role of biological micelles formed intraneuronally can be decisive in this regard and promote structural conversions into alternative conformations of TDP-43 NTD that may resist oligomerisation, LLPS and inclusion formation, also affecting the full-length protein.

## 4. Conclusions

The results presented here show that small concentrations of SB3-10 are able to promote an alternative conformational state of the NTD of TDP-43, highlighting its high structural plasticity. This new conformational state exhibits a different secondary structure, hydrophobic packing, size and oligomerisation state from the fully native state, but presents a fold with cooperativity and conformational stability only slightly lower than those of the native state. The solution conditions explored here represent a valid strategy to stabilize the NTD of TDP-43 as a monomer and this may be helpful to gain a better understanding of TDP-43 biological functions and its role in ALS and FTLD pathology. Furthermore, we speculate that the alternative NTD conformation might also form in vivo since biological micelles are normally found in neurons. This may be of physiological relevance, as this distinct conformational state might act differently in terms of functional oligomerisation and pathological aggregation also in vivo, although this awaits experimental demonstration. It could be interesting to further investigate the role of micelles in the monomerization and conformational change of the TDP-43 NTD and to explore if they are able to affect also the full-length protein. Our ability to stabilize TDP-43 in the monomer state is achievable and its employment for studies and approaches in vivo could be useful for future scientific applications.

## 5. Materials and Methods

### 5.1. Chemicals

SB3-10, acrylamide and ThT were from Sigma-Aldrich (St. Louis, MO, USA). DTT was from Thermo Fisher Scientific (Waltham, MA, USA).

### 5.2. Gene Cloning, Expression and Purification

Gene cloning, expression and purification of TDP-43 NTD and the Cys50Ser (C50S) single-point mutant were performed as previously described [24]. The purified protein contained 77 residues and the MHHHHHHSSGVDLGTENLYFQS sequence fused to the N-terminus for a total of 99 residues. It was maintained at 1.6–3.0 mg/mL (150–270 µM) in 5 mM sodium phosphate buffer, 50 mM NaCl, 1 mM DTT, pH 7.4, −20 °C. Protein purity was checked with SDS page. Protein concentration was measured using a SHIMADZU UV-1900 UV-Vis spectrophotometer at a wavelength of 280 nm with an extinction coefficient (ε_280_) of 12,950 M^−1^ cm^−1^.

### 5.3. Far-UV Circular Dichroism Spectroscopy

The TDP-43 NTD sample was centrifuged at 18,000× *g* for 15 min, 4 °C, and diluted to prepare two samples containing 0.5 mg/mL (45 µM) NTD, in 5 mM sodium phosphate buffer, 50 mM NaCl, 1 mM DTT, pH 7.4, one in the absence and the other in the presence of 3.0% (*w*/*v*) SB3-10. Spectra were acquired at 25 °C in the far-UV between 190 and 260 nm using a 0.1 mm path length cell on a Jasco J-810 spectropolarimeter (Tokyo, Japan) equipped with a thermostated cell holder attached to a Thermo Haake C25P water bath (Karlsruhe, Germany). Spectra were then blank subtracted, truncated when the high tension (HT) signal was higher than 700 V and normalized to mean residue ellipticity using
$$\left[\theta \right]=\frac{\theta }{\left(\frac{10\;\times \;N\;\mathrm{residues}\;\times \;\mathrm{optical}\;\mathrm{path}\;\times \;\mathrm{concentration}}{\mathrm{molecular}\;\mathrm{weight}}\right)}$$
where [*θ*] is the mean residue ellipticity in deg cm^2^ dmol^−1^, *θ* is the ellipticity in mdeg, the optical path is in cm, concentration is in g/L, and the molecular weight is in g/mol.

### 5.4. Fluorescence Spectroscopy

The TDP-43 NTD sample was centrifuged at 18,000× *g* for 15 min, at 4 °C, and two samples containing 0.5 mg/mL (45µM) NTD, in 5 mM sodium phosphate buffer, 50 mM NaCl and 1 mM DTT, pH 7.4, were prepared, one in the absence and the other in the presence of 3.0% SB3-10. Fluorescence spectra were acquired at 25 °C from 290 to 500 nm (excitation at 280 nm) using a 3 × 3 mm black wall quartz cell cuvette on an Agilent Cary Eclipse spectrofluorimeter (Agilent Technologies, Santa Clara, CA, USA) equipped with a thermostated cell holder attached to an Agilent PCB 1500 water Peltier system (Agilent Technologies, Santa Clara, CA, USA). Excitation and emission slits were 5 nm. Spectra were then blank subtracted.

### 5.5. Acrylamide Quenching Experiment

The TDP-43 NTD sample was centrifuged at 18,000× *g* for 15 min, at 4 °C, and two 1 mL samples containing 0.05 mg/mL (4.5 µM) NTD, in 5 mM sodium phosphate buffer, 50 mM NaCl and 1 mM DTT, pH 7.4, were prepared, one in the absence and the other in the presence of 3.0% SB3-10. For each sample, a first fluorescence spectrum was acquired at 25 °C from 300 to 420 nm (excitation at 280 nm) using a 10 × 4 mm quartz cuvette under magnetic stirring on an Agilent Cary Eclipse spectrofluorimeter (Agilent Technologies, Santa Clara, CA, USA) equipped with a thermostated cell holder attached to an Agilent PCB 1500 water Peltier system. Excitation and emission slits were 5 nm. Then, 5 µL of 1 M acrylamide was added directly to the cuvette and another spectrum was recorded. This step was repeated 15 times. For each recorded spectrum, the total fluorescence was corrected to take account of the dilution with the acrylamide solution. The quenching of acrylamide in the 0.0% SB3-10 sample was analyzed with the Stern-Volmer equation
$$\frac{F_{0}}{F}=1+K_{SV}\left[Q\right]$$
where *F*_0_ and *F* are the integrated fluorescence intensity areas at 300–400 nm in the absence and presence of acrylamide, respectively, *K_SV_* is the Stern-Volmer constant and [*Q*] is the concentration of the quencher (acrylamide) in the cuvette. The quenching of acrylamide in the 3.0% SB3-10 sample was instead analysed using
$$\frac{F_{0}}{F}=1+K_{SV}\left[Q\right]\;e^{K_{ST}\times \left[Q\right]}$$
where *K_ST_* is a constant that considers the static quenching caused by the binding of acrylamide to tryptophan residues.

### 5.6. Nuclear Magnetic Resonance Spectroscopy

NMR spectra were acquired with a Bruker Avance 500 MHz and a Bruker Avance Neo 700 MHz NMR spectrometers on ^13^C, ^15^N uniformly labelled TDP-43 NTD samples (250 µM) in 5 mM sodium phosphate, 50 mM NaCl, 3% (*w*/*v*) SB3-10, 1 mM DTT, 5% (*v*/*v*) D_2_O. For reference purposes, trimethylsilylpropanoic acid (TSP) was added [40]. For backbone assignment a series of 2D HSQC and 3D HNCA, HN(CO)CA, HNCACB, HNCO, and HN(CA)CO spectra were acquired at 25 °C and 17 °C with 2048 × 256, 2048 × 50 × 96, 2048 × 50 × 96, 2048 × 50 × 128, 2048 × 50 × 72, 2048 × 50 × 72 complex points, respectively. The spectral widths were 13.8 ppm (^1^H), 35 ppm (^15^N) and 30 ppm (^13^C in HNCA and HN(CO)CA), 16 ppm (^13^C in HNCO and HN(CA)CO), 80 ppm (^13^C in HNCACB). Secondary structures of individual residues were assessed by chemical shift index (CSI) [27], Talos+ [26] and Secondary Structure Propensity (SSP) [28]. According to Wishart [27], helix (H) or ^®^-strand (E) were indicated in Appendix A only if present a consensus for CSI for Cα, Cβ, C^®^, and CO. A series of 2D HSQC spectra were acquired at different temperature values (290–312K every 2 K). 2D ^15^N{^1^H} NOE measurements were also performed using a 700 MHz spectrometer at 25 °C and 17 °C, with a 5 s relaxation or saturation delay, 256 t_1_ increments of 2048 complex data points, and 48 scans/t_1_. The spectra were processed with Topspin 3.5 (Bruker Biospin) and analysed in Sparky [41]. NOE values were calculated as the ratio of peak height with and without saturation.

### 5.7. Dynamic Light Scattering

The TDP-43 NTD sample was centrifuged at 18,000× *g* for 15 min, at 4 °C, and filtered with Whatman Anotop 0.02 µm cut-off filters. Two samples containing 1.35 mg/mL (121 µM) NTD, in 5 mM sodium phosphate buffer, 50 mM NaCl and 1 mM DTT, pH 7.4, were prepared, one in the absence and the other in the presence of 3.0% SB3-10. Their size distributions (distribution of apparent hydrodynamic diameter by light scattering intensity) were then acquired on a Malvern Panalytical Zetasizer Nano S DLS device (Malvern Panalytical, Malvern, UK), thermostated at 25 °C with a Peltier temperature controller using a 3 × 3 mm black wall quartz cell cuvette. The refractive index and viscosity, acquired using a 2WAJ ABBE bench refractometer from Optika Microscopes (Bergamo, Italy) and a Viscoball viscometer (Fungilab, Barcelona, Spain), were 1.331 and 0.8998 cp for the 0.0% SB3-10 sample and 1.338 and 1.0530 cp for the 3.0% SB3-10 sample, respectively. The measurements were acquired with the cell position 4.20 and attenuator index 10. The light scattering intensity was also measured for a blank sample containing 3.0% SB3-10 without protein and was found to be negligible relative to the corresponding sample with protein, ruling out that the apparent hydrodynamic diameter measured for the protein sample under these conditions is affected by SB3-10 micelles. In another experiment, the same buffer solutions containing SB3-10 concentrations ranging from 0.0 to 3.0%, in the absence of protein were also analysed. Their light scattering intensities were then plotted as a function of SB3-10 concentration to monitor micelle formation.

### 5.8. Analytical SEC

The mother solution containing TDP-43 NTD was centrifuged at 18,000× *g* for 15 min, at 4 °C, and two samples containing 0.5 mg/mL (45µM) NTD, in 5 mM sodium phosphate buffer, 50 mM NaCl and 1 mM DTT, pH 7.4, were prepared, one in the absence and the other in the presence of 3.0% SB3-10. 100 µL of NTD sample were loaded in a Superdex^®^ 200 Increase 10/300 GL column (GE Healthcare, Chicago, IL, USA) pre-equilibrated at 4 °C with the corresponding protein buffer and run using an Akta Pure 25L System (GE Healthcare, Chicago, IL, USA). The experimental *V_e_* was determined as the volume of the buffer passed through the column between sample injection and the point of highest absorbance at 280 nm. A calibration curve was determined by loading protein standards (100 µL) of known molecular weight (MW) separately, such as thyroglobulin (669 kDa), apoferritin (443 kDa), alcohol dehydrogenase (150 kDa), albumin (66 kDa), carbonic anhydrase (29 kDa) and α-lactalbumin (14.2 kDa), and obtaining their experimental *V_e_*. A plot of *V_e_* versus log[MW] was obtained with the 6 standard data points and fitted to a linear regression curve. This calibration curve was then used to interpolate the experimental *V_e_* (y axis) of NTD samples and to obtain the corresponding MW (x axis) and, thus, their oligomeric states. The MW values determined with this approach are valid only for globular proteins. For the NTD sample in 3.0% SB3-10, the calibration curve was re-determined mathematically to account for the differences between the hydrodynamic radii of folded and pre-molten globule states [42]. Indeed, the relationships between the hydrodynamic radii of a folded protein ($R_{h}^{N}$), or a pre-molten globule ($R_{h}^{\mathrm{PMG}}$), and the MW in Da, can be obtained using previously published equations [41]. The experimental *V_e_* obtained for the NTD sample in 3.0% SB3-10 was interpolated into the re-determined calibration curve to obtain its MW.

### 5.9. Förster Resonance Energy Transfer (FRET)

The C50S mutant of TDP-43 NTD (containing only Cys39) was centrifuged at 18,000× *g* for 15 min, 4 °C, and diluted to prepare two samples containing 0.5 mg/mL (45 µM) C50S NTD, in 5 mM sodium phosphate buffer, 50 mM NaCl, 1 mM DTT, pH 7.4. The C50S NTD samples were labelled using the thiol-reactive probes 1,5-IAEDANS as a donor (D) and 6-IAF as an acceptor (A), respectively. Both fluorescent probes (Thermo Fisher Scientific, Waltham, MA, USA) were dissolved in dimethylformamide (DMF) at 30 mM concentration and added separately to the two C50S samples to a final probe:protein molar ratio of 10:1, for 22 h at 20 °C, under gentle and constant shaking. Then, gravity chromatography was performed to remove the excess of the unreacted probes using 6 mL of G-15 resin (Pharmacia, Uppsala, Sweden), previously equilibrated with 5 mM sodium phosphate buffer, 50 mM NaCl, pH 7.4.Labelledd protein fractions were collected and the concentration of the probe in each fraction was measured on a SHIMADZU UV-1900 UV-Vis spectrophotometer using ε_336_ = 5700 M^−1^cm^−1^ and ε_491_ = 8200 M^−1^cm^−1^ for 1,5-IAEDANS and 6-IAF, respectively. The C50S NTD concentration was determined using ε_280_ = 12,950 M^−1^cm^−1^ after subtraction of the absorbance contribution of the probes at 280 nm. Fractions where the probe:protein ratio was 1:1 were pooled and concentrated using centrifugal filter devices with a 3 kDa molecular weight cut-off (MWCO) cellulose membrane (Millipore, Burlington, MA, USA).

Two samples containing 0.1 mg/mL (9 µM) C50S NTD labelled with D and A, respectively, were prepared in 5 mM sodium phosphate buffer, 50 mM NaCl, 1 mM DTT and 3% SB3-10, pH 7.4, and mixed at 1:1 molar ratio (D:A). Two samples containing only C50S NTD labelled with D and C50S NTD labelled with A at the concentration of 0.05 mg/mL (4.5 µM) were also prepared. Fluorescence spectra of the various samples were acquired at 25 °C from 400 to 650 nm (excitation at 336 nm) using a 10 × 2 mm quartz cuvette on an Agilent Cary Eclipse spectrofluorimeter (Agilent Technologies, Santa Clara, CA, USA) equipped with a thermostated cell holder attached to an Agilent PCB 1500 water Peltier system. Excitation and emission slits were 5 nm. Spectra were then blank-subtracted.

### 5.10. SB3-10 Titration on NTD

The TDP-43 NTD sample was centrifuged at 18,000× *g* for 15 min, at 4 °C. The experiment was performed by preparing 16 samples containing NTD at the concentration of 0.5 mg/mL (45µM), in 5 mM sodium phosphate buffer, 50 mM NaCl and 1 mM DTT, pH 7.4, and SB3-10 concentrations ranging from 0.0 to 3.0%. Fluorescence spectra were acquired at 25 °C from 290 to 500 nm (excitation at 280 nm) using a 3 × 3 mm black wall quartz cell cuvette on an Agilent Cary Eclipse spectrofluorimeter (Agilent Technologies, Santa Clara, CA, USA) equipped with a thermostated cell holder attached to an Agilent PCB 1500 water Peltier system. Excitation and emission slits were 5 nm. Spectra were then subtracted from blanks containing only buffers. For each spectrum, COM was calculated according to
$$\mathrm{COM}=\frac{\sum_{i}F_{i}}{\sum_{i}v_{i}\cdot F_{i}}$$
where *F_i_* is the fluorescence emission at a wavenumber of *ν_i_*. The resulting COM values were then plotted as a function of SB3-10 concentration and fitted using the model edited by Santoro and Bolen [31].

### 5.11. Guanidine Hydrochloride (GdnHCl)-Induced Denaturation

The TDP-43 NTD sample was centrifuged at 18,000× *g* for 15 min, at 4 °C. The experiment was performed by preparing 30 samples containing NTD at the concentration of 0.05 mg/mL (4.5 µL), in 5 mM sodium phosphate buffer, 50 mM NaCl and 1 mM, pH 7.4, and GdnHCl concentrations ranging from 0.0 to 4.9 M. In a second experiment, 30 additional samples were prepared in the presence of 3.0% SB3-10. Fluorescence spectra were acquired at 25 °C from 290 to 500 nm (excitation at 280 nm) using a 10 × 2 mm quartz cuvette on an Agilent Cary Eclipse spectrofluorimeter (Agilent Technologies, Santa Clara, CA, USA), equipped with a thermostated cell holder attached to an Agilent PCB 1500 water Peltier system. Excitation and emission slits were 5 nm. Spectra were then subtracted from blanks containing only buffers. For each spectrum, the ratio between the sum of the fluorescence in bands of 10 nm in the post-transition region and the pre-transition region was calculated. These specific bands were chosen to contain the NTD fluorescence peak observed at 0.0 M GdnHCl and 4.9 M GdnHCl for the pre-transition and the post-transition, respectively. The resulting fluorescence ratios were then plotted as a function of the GdnHCl concentration and the traces obtained were then fitted to the model edited by Santoro and Bolen [31].

### 5.12. Aggregation Kinetics Using DLS, Far-UV CD and Intrinsic Fluorescence Spectroscopies

The TDP-43 NTD sample was centrifuged at 18,000× *g* for 15 min, at 4 °C. Samples containing NTD at the concentration of 0.5 mg/mL (45 µM), in 5 mM sodium phosphate buffer, 50 mM NaCl, 1 mM DTT, pH 7.4, 25 °C were prepared using different SB3-10 concentrations (0.0, 0.2, 0.4, 0.6, 1.8 and 3.0%). One sample was obtained by treating the protein, initially in 0.0% SB3-10, with 1.8% SB3-10 for 1 h and then diluting it down to 0.6% SB3-10. Samples were analyzed every 30 min for the first 2 h, then hourly up to 6 h and the next day at 24 h. Their size distributions were acquired at 25 °C using a 3 × 3 mm black wall quartz cell cuvette on a Malvern Panalytical Zetasizer Nano S DLS device (Malvern, Worcestershire, UK), thermostated at 25 °C with a Peltier temperature controller. The refractive index and viscosity set on the instrument were changed according to the SB3-10 content of the sample. The measurements were acquired with the cell position 4.20 and attenuator index 10. The light scattering intensity percentage of the soluble protein peak was then calculated for each sample and plotted as a function of time. Their far-UV CD spectra were acquired at 25 °C between 190 and 260 nm using a 0.1 mm path length cell on the same Jasco J-810 spectropolarimeter described above. Spectra were then blank subtracted, masked when the high tension (HT) signal was higher than 700 V and normalized to mean residue ellipticity ([*θ*]). Their fluorescence spectra were acquired at 25 °C from 290 to 500 nm (excitation at 280 nm) using a 10 × 2 mm quartz cuvette on an Agilent Cary Eclipse spectrofluorimeter (Agilent Technologies, Santa Clara, CA, USA), equipped with a thermostated cell holder attached to an Agilent PCB 1500 water Peltier system. Excitation and emission slits were 5 nm. Spectra were then blank subtracted.

### 5.13. Thioflavin T (ThT) Assay

TDP-43 NTD samples were prepared at various SB3-10 concentrations, as described in the previous subsection and then incubated for 6 h at 25 °C, while the sample in 3.0% SB3-10 was incubated also for 24, 48, 72 and 96 h. 100 µL of buffer or protein sample were then mixed with 400 µL of 25 µM ThT. Fluorescence spectra were acquired at 25 °C from 450 to 600 nm (excitation at 440 nm) using a 10 × 2 mm quartz cuvette on an Agilent Cary Eclipse spectrofluorimeter (Agilent Technologies, Santa Clara, CA, USA) equipped with a thermostated cell holder attached to an Agilent PCB 1500 water Peltier system. Excitation and emissions slits were 5 and 10 nm, respectively. All spectra were blank subtracted (using only PBS as a blank). The ratio *F*_0_/*F* was then calculated for each sample, where *F* and *F*_0_ are the blank-subtracted fluorescence values at 485 nm of ThT+protein and free ThT, respectively. An over 5-fold ThT fluorescence increase was considered to be diagnostic for amyloid [33,34,43,44].

### 5.14. Statistical Analysis

Data were expressed as means ± SEM. Data pairs were compared using the Student’s *t*-test and the resulting *p* values were indicated in the text for each comparison. A *p* value < 0.05 was considered to be statistically significant.

## Figures and Tables

**Figure 1 molecules-27-04309-f001:**
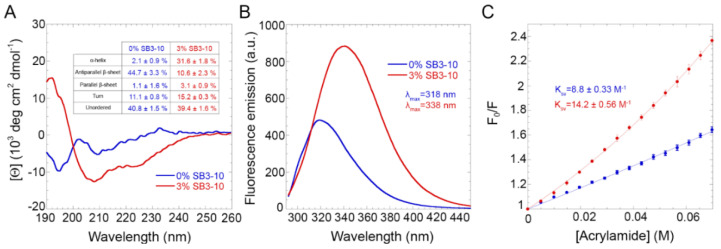
Structural characterization of TDP-43 NTD. Far-UV CD spectra (**A**), Intrinsic fluorescence spectra (**B**) and Stern-Volmer assays (**C**) of TDP-43 NTD in 0.0% (blue) and 3.0% (red) SB3-10. All SB3-10 percentages are *w*/*v*. The writings inside the panels indicate the values obtained for the analysed parameters.

**Figure 2 molecules-27-04309-f002:**
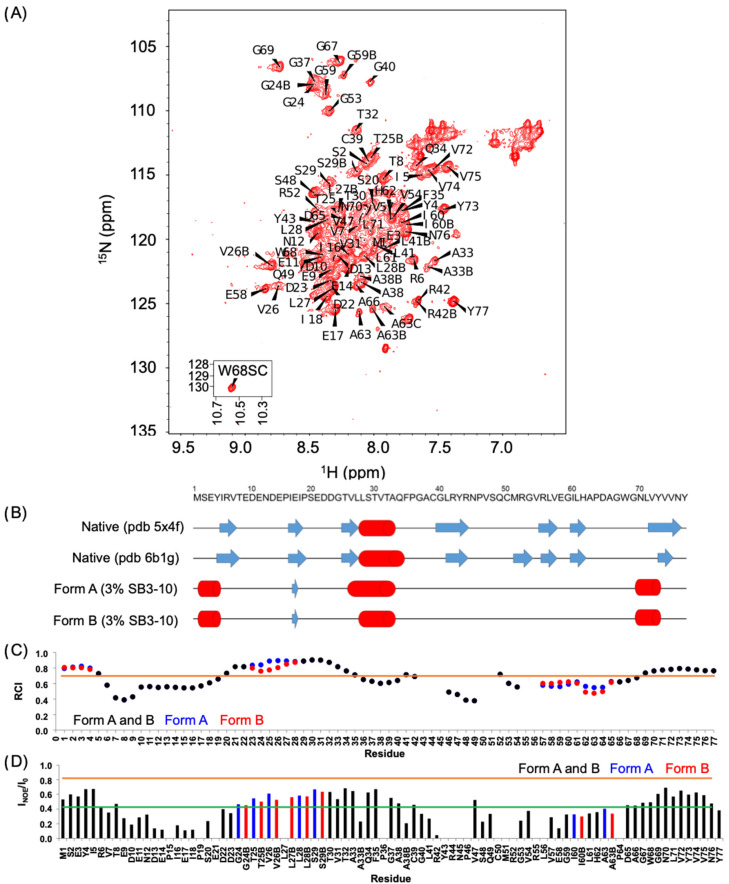
NMR characterization of TDP-43 NTD in 3% SB3-10. (**A**) ^1^H-^15^N HSQC spectrum of TDP-43 NTD in 3% (*w*/*v*) SB3-10 at 25 °C, recorded at 700 MHz. (**B**) Comparison of the secondary structure of native NTD without SB3-10 (pdb id: 5x4f and 6b1g) and of the alternative conformation in 3% SB3-10 reporting the consensus analysis of Talos+, CSI and SSP for forms (**A**) and (**B**). Blue arrows indicate extended structures (β-strands) and red cylinders stand for α-helices. (**C**) Random coil index (RCI) values as predicted by Talos+. The black, blue and red bars correspond to both forms, form (**A**) and form (**B**), respectively. RCI values correspond to order parameters (S^2^). Residues with S^2^ ≤ 0.7 (orange line) are considered too flexible to produce reliable torsion angles [26]. (**D**) ^15^N{^1^H} NOE ratios at 25 °C. Colour codes as in panel (**C**). The green and orange lines indicate the average NOE value and the NOE value expected for a rigid structure (0.82) [30], respectively.

**Figure 3 molecules-27-04309-f003:**
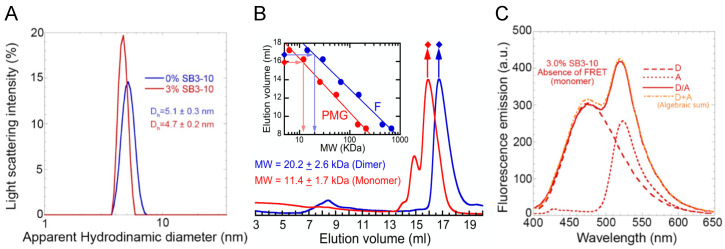
Oligomeric state characterization of TDP-43 NTD. DLS distributions (**A**), analytical size exclusion chromatography (SEC) profiles (**B**) and FRET analyses (**C**) of TDP-43 NTD in 0.0% (blue) and 3.0% (red) SB3-10. All SB3-10 percentages are *w*/*v*. In panel (**B**) the inset shows the calibration curve obtained with proteins of known masses (blue) and re-determined mathematically for pre-molten globule states, as described in Methods (red). In panel (**C**) the spectra refer to NTD-D (dashed line), NTD-A (dotted line), NTD-D/NTD-A (continuous line) and the algebraic sum of the first two spectra (mixed orange dotted/dashed line). The writings inside the panels indicate the values obtained for the analysed parameters.

**Figure 4 molecules-27-04309-f004:**
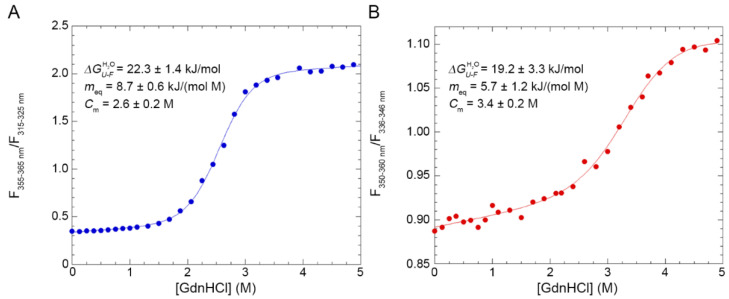
GdnHCl-induced denaturation curves of TDP-43 NTD. The curves refer to TDP-43 NTD in 0.0% (**A**) and 3.0% (**B**) SB3-10. All SB3-10 percentages are *w*/*v*. The ratio between the fluorescence values observed at the indicated wavelength windows, in which fluorescence emission was marked at high and low GdnHCl concentrations, respectively, was obtained at each GdnHCl concentration and plotted as a function of the GdnHCl concentration. Data were analysed with a best-fitting procedure using the two-state model equation [31]. The writings inside the panels indicate the values obtained for the analysed parameters.

**Figure 5 molecules-27-04309-f005:**
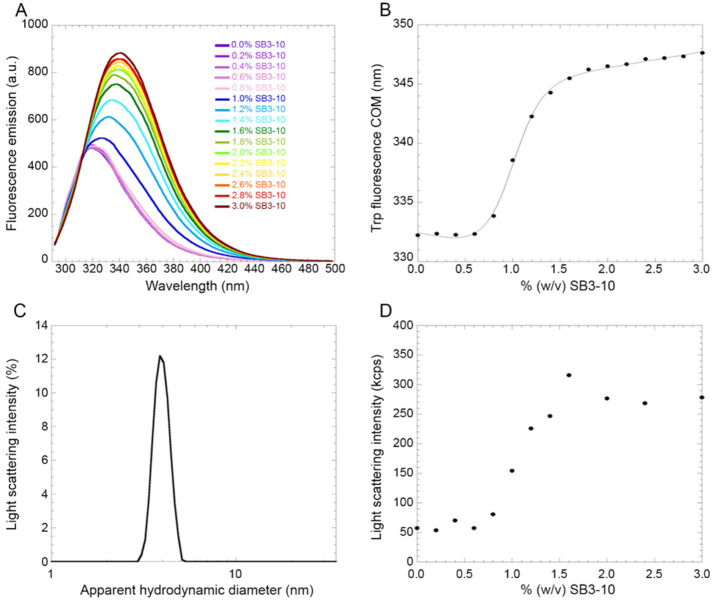
SB3-10-induced denaturation curve of TDP-43 NTD. (**A**) Fluorescence spectra of TDP-43 NTD (excitation 280 nm) at the indicated SB3-10 percentages (*w*/*v*). (**B**) Plot of spectral COM versus SB3-10 percentage (*w*/*v*). Data were analysed with a best-fitting procedure using the two-state model equation [31]. (**C)** Representative DLS distribution of a solution containing 3.0% (*w*/*v*) SB3-10 in the absence of protein. Similar distributions were obtained at all SB3-10 percentages analysed here, using a fixed attenuator and measurement position on the instrument. (**D**) Plot of light scattering intensity as a function of the SB3-10 percentage (*w*/*v*) in the absence of protein, using a fixed attenuator and measurement position on the instrument.

**Figure 6 molecules-27-04309-f006:**
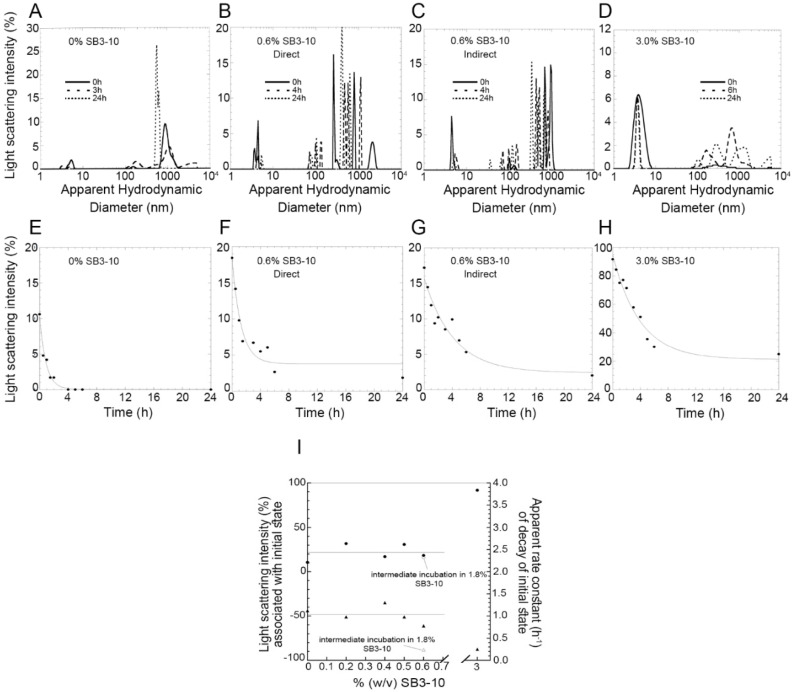
Aggregation kinetics of TDP-43 NTD at different SB3-10 percent concentrations monitored with DLS. (**A**–**D**) Size distributions of TDP-43 NTD (0.5 mg/mL, 45 µM) in 5 mM sodium phosphate buffer, 50 mM NaCl, 1 mM DTT, pH 7.4, 25 °C, in the presence of 0.0% (*w*/*v*) SB3-10 (**A**), 0.6% (*w*/*v*) SB3-10 (**B**,**C**), 3.0% (*w*/*v*) SB3-10 (**D**), after 0 h (continuous lines), 3–6 h (dashed lines) and 24 h (dotted lines). (**E**–**H**) Light scattering intensity associated with the initial state (low-molecular weight monomer/dimer/oligomer) of TDP-43 NTD versus time. Conditions as in the corresponding top panels. In panels (**C**), (**G**) TDP-43 NTD was pre-incubated in 1.8% (*w*/*v*) SB3-10 for 1 h and then diluted to 0.6% (*w*/*v*) SB3-10 at the same final conditions as in panels (**B**,**F**). (**I**) Light scattering intensity associated with initial state (circles), and apparent rate constant of decay of initial state (triangles), versus SB3-10 percentage (*w*/*v*). The empty symbols refer to the sample pre-incubated in 1.8% (*w*/*v*) SB3-10 for 1 h and then diluted to 0.6% (*w*/*v*) SB3-10. Horizontal lines indicate mean values. All data were acquired using a fixed attenuator and measurement position on the instrument.

**Figure 7 molecules-27-04309-f007:**
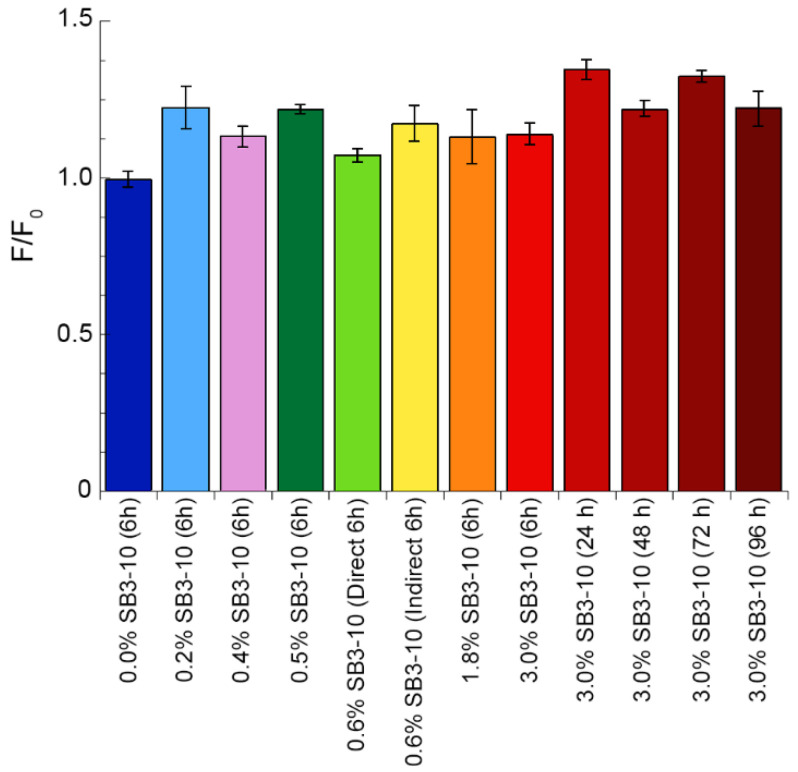
Thioflavin T (ThT) assay of TDP-43 NTD at different SB3-10 percent concentrations. TDP-43 NTD (0.5 mg/mL, 45 µM) in 5 mM sodium phosphate buffer, 50 mM NaCl, 1 mM DTT, pH 7.4, 25 °C, in the presence of the indicated SB3-10 percent concentrations (*w*/*v*) and time points. ThT assay was performed with excitation at 440 nm and fluorescence emission at 450–600 nm and reported as the ratio between ThT fluorescence at 485 nm in the presence (*F*) and absence (*F*_0_) of TDP-43 NTD (n = 3, mean ± SEM). Differences were in some cases statistically significant relative to the first sample in 0.0% SB3-10 at 6 h but considered to be non-relevant because an increase of the *F*/*F*_0_ value is expected to be at least 5-fold for amyloid (see text for details).

**Figure 8 molecules-27-04309-f008:**
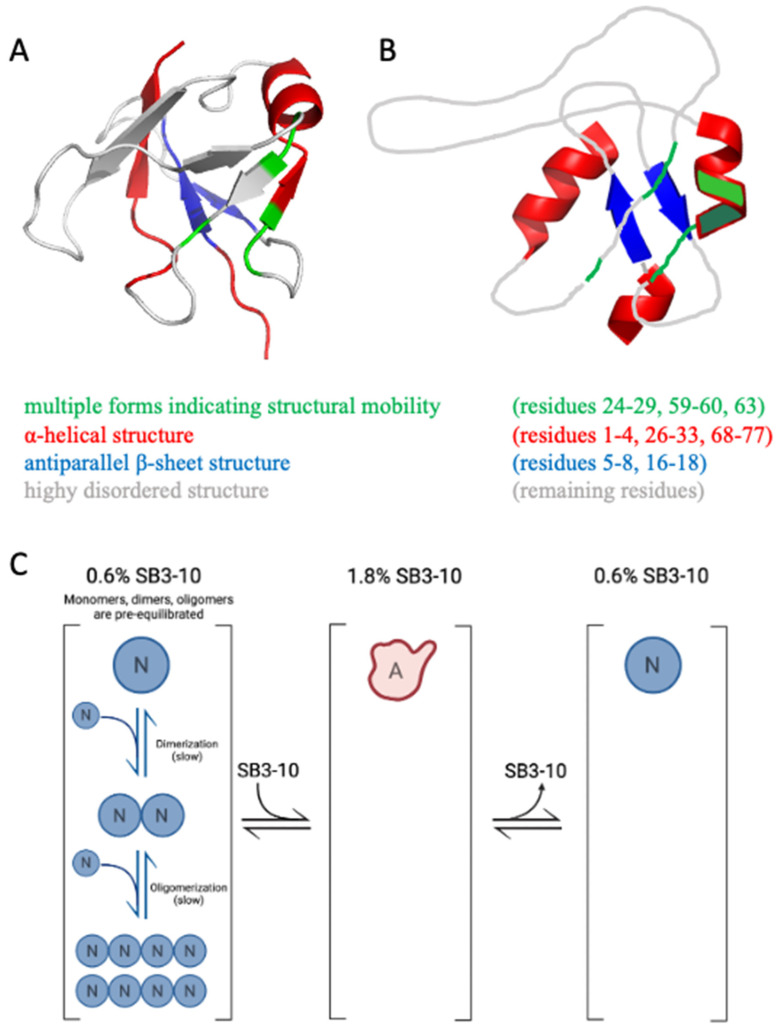
(**A**) NMR structure of monomeric native TDP-43 NTD (PDB ID 5x4f) colour coded to illustrate residues adopting multiple forms (green), α-helical structure (red), antiparallel β-sheet structure (blue) and highly disordered structure or loops (grey) in the alternative conformation studied here in 3% (*w*/*v*) SB3-10. (**B**) Proposed model of the same alternative conformation of TDP-43 NTD in 3% (*w*/*v*) SB3-10, colour coded as in panel (**A**). Orientation of helices, β-sheet and loops in the images is arbitrary as the relative orientation of secondary structure elements has not been attempted. (**C**) Scheme showing that TDP-43 NTD has a low propensity to dimerise/oligomerise in the alternative conformation. N and A indicate the native and alternative conformational states, respectively. N monomers, dimers and oligomers are initially pre-equilibrated (slow conversions). A is a monomer. The lower propensity of TDP-43 NTD to oligomerise and self-assemble is maintained when A is re-located in native conditions to form N (right) because the conversion to dimers and oligomers is slow.

## Data Availability

Not applicable.

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
