# Peer review of "Conversion of the Native N-Terminal Domain of TDP-43 into a Monomeric Alternative Fold with Lower Aggregation Propensity"

_molecules, 2022, doi:10.3390/molecules27134309_

Round 1
Reviewer 1 Report
In the present work, Matteo Moretti and colleagues show the high plasticity of TDP-43 NTD and identify strategies to monomerize TDP-43 NTD for methodological and biomedical applications.
In particular, by means of biochemical and biophysical investigations, the authors suggest that the NTD of TDP-43 has high plasticity as it forms a conformational state in the presence of small concentrations of SB3-10 that is distinct from the fully native state in terms of secondary structure, hydrophobic packing, size, and oligomerization state, although it presents a fold with cooperativity and conformational stability only slightly lower than those of the native state.
This is an interesting aspect in the field of protein biochemistry and deserves attention. However, there are some important issues that emerged in the paper and I would like to rise.
1- Why did you select Sulfobetaine 3-10 (SB3-10) as a detergent in this study?
2- Why did you select Cys50Ser (C50S) amount other single-point mutants of TDP-43 NTD in this study?
3- As some mutants of the TDP-43 NTD result in mislocalization to cytoplasm and formation of inclusions, why you don’t consider these mutants in your study.
2- How you can link your results come from the TDP-43 NTD analysis with post-translational modifications of TDP-43 and its in-vivo function?
2- Statistical analysis methods are not mentioned clearly, also please note that in each corresponding figure legend.
3- The main criticism of this study is the lack of suitable conclusions and remarks that could account for future scientific applications. Intense research is being conducted in many areas related to ALS, from basic science seeking the roots of the disease to therapy development to find effective treatments. However, translating such basic biochemical research data into the clinic has not been straightforward and has presented many scientific and regulatory challenges for scientists. In this regard, authors should interpret and discuss the significance of their findings in the specific section (conclusions and remarks).
Author Response
Author’s reply to review report 1
Reviewer’s point:
1- Why did you select Sulfobetaine 3-10 (SB3-10) as a detergent in this study?
Author’s reply:
This zwitterionic detergent was initially used to stabilize the protein domain during purification, as it is normally used in protein purification protocols. However, we soon realized that it led to a different conformational state of the protein domain and we therefore decided to use it more systematically to investigate the structural plasticity of the TDP-43 NTD. However, we missed reporting this explanation in the first version of the manuscript. This argument was therefore added at the beginning of the Results section (lines 107-111).
Reviewer’s point:
2- Why did you select Cys50Ser (C50S) amount other single-point mutants of TDP-43 NTD in this study?
Author’s reply:
We wanted to perform a FRET analysis using fluorescent cysteine-reactive probes (6-IAF and 1,5-IAEDANS). Since the NTD has two cysteine residues (at positions 39 and 50, respectively) we produced a mutant that contained only one cysteine residue (C50S) to be sure that the donor probe linked to a NTD molecule interacted only with the acceptor probe of a second NTD on the corresponding cysteine residue. This mutation is not pathological and was not found in familial ALS or FTLD-U patients. This protein variant had already been produced and studied (Jiang et al. 2017 Sci Rep, 7(1):6196) and no significant secondary structure variation was found with respect to the wild-type protein. Hence, we decided to produce it in order to perform the FRET experiment aforementioned.
Reviewer’s point:
3- As some mutants of the TDP-43 NTD result in mislocalization to cytoplasm and formation of inclusions, why you don’t consider these mutants in your study.
Author’s reply:
The aim of our study was to compare the two NTD conformations with the wild-type sequence. We are aware that some research groups studied the folding and functions of some TDP-43 NTD mutations like L27A, L28A and V31R (Mompeán et al. 2017 J Biol Chem, 292(28):11992-12006; Kumar et al. 2019 Biophys chem, 250:106174). However, the aim of our study was to identify the conditions to obtain the two NTD conformational states having the wild-type sequence and to compare them in order to find similarities and differences, so we deliberately chose not to include mutations.
Reviewer’s point:
4- How you can link your results come from the TDP-43 NTD analysis with post-translational modifications of TDP-43 and its in-vivo function?
Author’s reply:
We understand the reviewer’s point, which is highly pertinent because post-translational modifications (PTMs) are very important in TDP-43 biology. PTMs of TDP-43 involve (i) C-terminal truncation, (ii) hyperphosphorylation and (iii) polyubiquitination. None of them, surprisingly, involve the NTD of TDP-43 because (i) proteolysis occurs well beyond the C-terminal end of the NTD, (ii) all the identified phosphorylation sites are again beyond the CTD and (iii) the NTD has not any lysine residues, making it impossible for the cell to ubiquitinate it. Since our study involves an isolated NTD of TDP-43, PTMs are not relevant to this system.
Reviewer’s point:
5- Statistical analysis methods are not mentioned clearly, also please note that in each corresponding figure legend.
Author’s reply:
The reviewer is correct. We have indeed overlooked the statistical analysis in many graphs and data analyses. Pairs of values of the various parameters were compared using the Student’s t-test and the resulting p values were indicated in the text (lines 126, 138, 278, 284, 286, 363, 365, 375, 384, 385). In Figure 7, differences of the F/F0 ratio of ThT fluorescence were in some cases found to be statistically significant relative to the first sample in 0.0% SB3-10 at 6 h, but considered to be non-relevant because an increase of the F/F0 value is expected to be at least 5-fold for amyloid and was also lower than 1.4 in our measurements. This point has been added in the legend of Figure 7.
Statistical analysis was not performed when CD spectra, fluorescence spectra, NMR spectra, DLS distributions, etc were clearly different, as this is not usually used in spectroscopic analyses where the reproducibility of the recording is very high.
Reviewer’s point:
6- The main criticism of this study is the lack of suitable conclusions and remarks that could account for future scientific applications. Intense research is being conducted in many areas related to ALS, from basic science seeking the roots of the disease to therapy development to find effective treatments. However, translating such basic biochemical research data into the clinic has not been straightforward and has presented many scientific and regulatory challenges for scientists. In this regard, authors should interpret and discuss the significance of their findings in the specific section (conclusions and remarks).
Author’s reply:
We have created a new section Conclusions (lines 548-564), as recommended by the reviewer. We have reorganized the text of this section relative to the last paragraph of our previous Discussion section to make the applications of our results emerge more clearly. The applications are mainly for further studies in vivo and in vitro. Honestly, we are not sure that the results presented here have immediate applications in the clinics and for this reason we have not mentioned arguments on this point, that would have implied overselling our message.
Reviewer 2 Report
The authors present a well-designed and conducted study on the structural characteristics of folding intermediates of the TAR DNA-binding protein 43, TDP-43), and describe an additional folding intermediate containing some of the characteristics of the native protein, and slower folding and conformational stability, that could be responsible for the formation of cytoplasmatic inclusions observed ALS. This is an interesting study that could provide some light to the biochemical/physiological processes causatives of ALS.
I see no mayor corrections needed, only a few comments to make the manuscript easy to understand:
Line 67: “six-seven” not clear of the meaning, my guess is that there are six or seven (I count seven or eight in the Figure 2 B).
The terminology is confusing: TDP-43 is for the full-length protein and NTD the terminal domain. But sometimes the authors use the TDP-43 NTD, my assumption is that this is the N-terminal domain of the TAR DNA-binding protein, then according to their own definition, this should be simply refer as NTD in the manuscripp
A few minor spelling corrections are needed like:
Line 22: the Greek letter β is missing
Line 42: perhaps the “and” between polyubiquination and hyperphosphorylation could replace by a “,”
Author Response
Author’s reply to review report 2
Reviewer’s point:
1- Line 67: “six-seven” not clear of the meaning, my guess is that there are six or seven (I count seven or eight in the Figure 2 B).
Author’s reply:
The reviewer’s observation is correct and we have changed it (line 65).
Reviewer’s point:
2- The terminology is confusing: TDP-43 is for the full-length protein and NTD the terminal domain. But sometimes the authors use the TDP-43 NTD, my assumption is that this is the N-terminal domain of the TAR DNA-binding protein, then according to their own definition, this should be simply refer as NTD in the manuscripp
Author’s reply:
We realized that the reader may not be confident in the type of protein construct used in our analysis, particularly at the beginning of our description. For this reason, we have replaced “TDP-43 NTD” with “NTD of TDP-43” in lines 77, 93, 103. The reader will, by then, be familiar that the analysis is entirely on that protein domain and we mention freely “the NTD” or “the TDP-43 NTD” in later parts. In addition, using the protein name followed by the domain name is a common custom and there are several articles that refer to the N-terminal domain of TDP-43 as TDP-43 NTD (Afroz et al. 2017 Nat Commun, 8(1):45; Mompeán et al. 2017 J Biol Chem, 292(28):11992-12006; Wang et al. 2018 EMBO J, 37(5):e97452). We are confident that the description is clear in the revised form.
Reviewer’s point:
3- Line 22: the Greek letter β is missing
Author’s reply:
We changed it, as requested (line 21).
Reviewer’s point:
4- Line 42: perhaps the “and” between polyubiquination and hyperphosphorylation could replace by a “,”
Author’s reply:
Two “and” were initially present because “abnormal” was referred to both polyubiquitination and hyperphosphorylation (first “and”). However, the reviewer is correct and the sentence could actually be revised to a more comprehensible form. Hence, we changed the second “and” between hyperphosphorylation and partial proteolysis (line 40) with “as well as”. We hope that the revised statement is clearer.
Reviewer 3 Report
The manuscript describes a study combining a broad variety of approaches to study an alternative structure of TDP-43 NTD. The methods are chosen well, performed carefully, and described in sufficient details. However, I have some concerns about presenting and interpreting the results:
(1) Assigned chemical shifts of the new form(s) of NTD must be deposited in BMRB and accession number provided in the paper.
(2) Presenting the secondary structure of the new forms is somewhat inconsistent. Fig. 2 shows Talos+ analysis, predicting only one short beta strand, whereas two antiparalel strands are mentioned in the text (l. 169, l. 460), presumably based on the CSI analysis. This should be clarified.
(3) The quantitative data used for the secondary structure prediction should be presented. It includes the observed chemical shifts but also the reference random coil values. Reference to the source of random-coil values used in the study should be provided. Secondary Structure Propensity (SSP) values can be also provided as an alternative descriptor.
(4) It is not clear from the description why the native ("N") state prepared from the "A" form does not oligomerizes if Fig. 8 depicts all conversions as reversible equilibria. Even if the conditions temporarily promote monomeriation of the alternative conformation and disrupt intermolecular interactions (l. 499), why the equilibria in left and right brackets in Fig. 8C differ when the conditions appear to be identical?
(5) Presenting the secondary-structure model as Fig. 8B is somewhat misleading as the mutual orientation of the secondary structure elements is unknown. Have the authors tried to determine the orientation e.g. by measuring residual dipolar couplings? Considering the proposed role of micelles, partial alignment of the sample in bicelles would be interesting. But using a more readily available alignment (e.g. Pf1 phage) might be sufficient. One-bond H-N RDC can be measured in several hours using simple IPAP experiments.
Author Response
Author’s reply to review report 3
Reviewer’s point:
1- Assigned chemical shifts of the new form(s) of NTD must be deposited in BMRB and accession number provided in the paper.
Author’s reply:
As suggested by the reviewer, we have deposited the NMR assignment for both forms A and B with accession number 51492. We have also added the following sentence:
“1H, 13C and 15N chemical shifts have been deposited in the BioMagResBank (http://www.bmrb.wisc.edu) under the BMRB accession number 51492.” (lines 165-167).
Reviewer’s point:
2- Presenting the secondary structure of the new forms is somewhat inconsistent. Fig. 2 shows Talos+ analysis, predicting only one short beta strand, whereas two antiparalel strands are mentioned in the text (l. 169, l. 460), presumably based on the CSI analysis. This should be clarified.
Author’s reply:
As described in the text (lines 175-180) both Talos+, CSI and SSP (added) identify a short b-strand (16-18) and we have decided to present in Figure 2 the consensus secondary structure only (as now specified in the caption of Figure 2). We observe, however, that CSI, as well as SSP although with a low score, indicates residues 5,7 in extended conformation in agreement with the secondary structure of native TDP-43. In addition, Talos+ suggests a highly dynamic motion in the region. As suggested by the reviewer, we have rephrased the sentence in the manuscript (lines 175-180).
Reviewer’s point:
3- The quantitative data used for the secondary structure prediction should be presented. It includes the observed chemical shifts but also the reference random coil values. Reference to the source of random-coil values used in the study should be provided. Secondary Structure Propensity (SSP) values can be also provided as an alternative descriptor.
Author’s reply:
We thank the reviewer for the suggestion of an alternative analysis method based on chemical shift values. We have reported in the manuscript the results using SSP that confirm our previous analysis (lines 177-178 and Figure S4. We have specified in the Materials and Methods section how the helix and strand conformations were predicted following the classical CSI approach of Wishart (Wishart et al. 1994 J Biomol NMR 44, 213-223).
Reviewer’s point:
4- It is not clear from the description why the native ("N") state prepared from the "A" form does not oligomerizes if Fig. 8 depicts all conversions as reversible equilibria. Even if the conditions temporarily promote monomerisation of the alternative conformation and disrupt intermolecular interactions (l. 499), why the equilibria in left and right brackets in Fig. 8C differ when the conditions appear to be identical?
Author’s reply:
The reviewer’s point is correct and we admit that our figure and text description were misleading and unclear. The physiological N state (Fig. 8, left) consists of a high population of dimers in a slow pre-equilibrium with a low population of monomers and this makes the oligomerization reaction kinetically favorable. Monomers, dimers and oligomers exist initially in a pre-equilibrium where the mutual conversions are slow (Fig. 8, left). This condition is favourable for aggregation as dimers and oligomers are present already. The alternative state has a very low propensity to oligomerize (Fig. 8, centre), but such a lower propensity is maintained when the alternative state is re-located under native conditions to form the native state (Fig. 8, right), because the conversion to dimers and oligomers is slow. We have added this description in the text (Lines 527-533), in the legend to Figure 8 (lines 575-579) and we have also changed Figure 8 adding explanatory writings.
Reviewer’s point:
5- Presenting the secondary-structure model as Fig. 8B is somewhat misleading as the mutual orientation of the secondary structure elements is unknown. Have the authors tried to determine the orientation e.g. by measuring residual dipolar couplings? Considering the proposed role of micelles, partial alignment of the sample in bicelles would be interesting. But using a more readily available alignment (e.g. Pf1 phage) might be sufficient. One-bond H-N RDC can be measured in several hours using simple IPAP experiments.
Author’s reply:
We appreciate the reviewer’s point. The determination of the orientation of the secondary structure elements is not feasible within the short time requested for the revision. Due to this limitation we have clearly reported this point in the legend to Figure 8 (lines 572-573).